

# The recombinant BMP-2 loaded silk fibroin microspheres improved the bone phenotype of mild osteogenesis imperfecta mice

Ting Fu[*], Yi Liu[*], Zihan Wang, Yaqing Jing, Yuxia Zhao, Chenyi Shao, Zhe Lv and Guang Li

Department of Genetics, School of Basic Medical Sciences, Tianjin Medical University, Tianjin, China
[*] These authors contributed equally to this work.

Corresponding author
Guang Li, lig@tmu.edu.cn

## ABSTRACT

Osteogenesis imperfecta (OI) is an inherited congenital disorder, characterized primarily by decreased bone mass and increased bone fragility. Bone morphogenetic protein-2 (BMP-2) is a potent cytokine capable of stimulating bone formation, however, its rapid degradation and unanticipated *in vivo* effects restrict its application. The sustained release characteristic of silk fibroin (SF) microspheres may potentially address the aforementioned challenges, nevertheless they have not previously been tested in OI treatment. In the current investigation, recombinant BMP-2 (rBMP-2) loaded SF (rBMP-2/SF) microspheres-based release carriers were prepared by physical adsorption. The SF microparticles were characterized by scanning electron microscopy (SEM) and were investigated for their cytotoxicity behavior as well as the release profile of rBMP-2. The rBMP-2/SF microspheres were administered via femoral intramedullary injection to two genotypes of OI-modeled mice daily for two weeks. The femoral microstructure and histological performance of OI mice were evaluated 2 weeks later. The findings suggested that rBMP-2/SF spheres with a rough surface and excellent cytocompatibility demonstrated an initial rapid release within the first three days ($22.15 \pm 2.88\%$ of the loaded factor), followed by a transition to a slower and more consistent release rate, that persisted until the 15th day in an *in vitro* setting. The factor released from rBMP-2/SF particles exhibited favorable osteoinductive activity. Infusion of rBMP-2/SF microspheres, as opposed to blank SF spheres or rBMP-2 monotherapy, resulted in a noteworthy enhancement of femoral microstructure and promoted bone formation in OI-modeled mice. This research may offer a new therapeutic approach and insight into the management of OI. However, further investigation is required to determine the systematic safety and efficacy of rBMP-2/SF microspheres therapy for OI.

## INTRODUCTION

Osteogenesis imperfecta (OI) is an inherited disorder characterized by low bone mass and increased bone fragility (*Marom, Rabenhorst & Morello, 2020*; *Rossi, Lee & Marom, 2019*).

According to reports, the global incidence of OI is estimated to be around $1/20\,000 - 1/10\,000$. Although several disease-causing genes have been identified in OI patients, 85% to 90% of clinical cases were found to be associated with pathogenic variants in *COL1A1* or *COL1A2* (*Claeys et al., 2021*), which encode procollagen peptides $\alpha1$ or $\alpha2$ of type I collagen. The bone matrix is predominantly comprised of collagen protein, which renders the bone highly susceptible to the effects of OI. A mutation in either *COL1A1* or *COL1A2* genes frequently results in either quantitatively or structurally impaired fibrils of collagen, leading to compromised accumulation of mineral content and thereby reduced bone strength (*Jovanovic, Guterman-Ram & Marini, 2022*; *Folkestad et al., 2012*). According to Sillence's classification, individuals with collagen deficiency can be categorized into four subgroups based on the severity and manifestation of their symptoms (*Sillence, Rimoin & Danks, 1979*). The mild phenotype of OI type I can be attributed to reduced collagen production. Abnormal collagen structure is responsible for the development of lethal OI type II, severe OI type III with bone deformities, and moderately severe OI type IV.

Currently, the principal clinical treatments for OI are orthopedic surgery and anti-resorptive drugs (*Marini et al., 2017*; *Morello, 2018*). The implementation of a surgical approach has been shown to effectively restore both the anatomical structure and function of bones in patients experiencing fractures and bone malformations (*Harrington, Sochett & Howard, 2014*). The nitrogen-containing bisphosphonates (BPs) are the mainstay drugs for OI patients, particularly children (*Lindahl et al., 2016*). It has been demonstrated that BPs given intravenously or orally increase bone mass by inhibiting osteoclast activity or inducing osteoclast apoptosis (*Marom et al., 2016*). However, the long half-life of BPs raises the concern that long-term inhibition of bone absorption may impair bone elasticity (*Morello, 2018*).

Osteoblasts and osteoclasts are continuously involved in the remodeling of bone tissue (*Kim et al., 2020*). Due to increased bone resorption and decreased bone production, several OI types show high bone turnover (*Marini & Dang, 2000*). Bone anabolic therapy may be more advantageous than anti-resorptive methods, especially in lowering the frequency of bone fractures. Following the treatment of teriparatide, sclerostin antibody, and transforming growth factor-$\beta$ (TGF-$\beta$) antibody, osteoblastic bone production has been observed to be enhanced in several kinds of OI (*Nicol et al., 2021*; *Sinder et al., 2014*; *Tauer, Abdullah & Rauch, 2019*). The utilization of multiple osteogenic inducing agents, including bone morphogenetic protein-2 (BMP-2) and Nell1, has been observed to yield favorable results in the treatment of OI (*Cheng et al., 2019*; *Liu et al., 2020*). However, the relatively rapid clearance and diffusion rates of these exogenous drugs or factors frequently necessitate high dosages and multiple administrations. Sustained release of drugs with the help of biomaterials can not only prolong their efficacy but also reduce dosing frequency and several potential adverse effects of systemic administration.

The mechanical properties, biocompatibility, and biodegradability of silk fibroin (SF) derived from natural silk have rendered it a highly versatile material for use in various biomedical applications (*DeBari et al., 2021*). The stable drug-carrying capacity of SF allows for the targeted delivery of therapeutic agents to specific sites, facilitating sustained and localized release at the site of disease (*Farokhi et al., 2020*; *Numata & Kaplan, 2010*).

By using embedding or adsorption techniques, microspheres can be created from SF, allowing the prolonged release of protein components or drugs. (*Zhang et al., 2019*). Moreover, SF exhibits a structural resemblance to type I collagen. The utilization of SF microspheres containing osteogenic factors, either independently or in conjunction with other biomaterials such as collagen and chitosan, has been implemented in the repairing of bone defects and the engineering of bone tissue (*Seong et al., 2020*; *Li et al., 2021*). Although there is currently no practical application in the treatment of OI.

In this study, recombinant BMP-2 (rBMP-2)-loaded SF (rBMP-2/SF) microspheres were prepared *via* physical adsorption. These microspheres were then administered daily *via* femoral intramedullary injection to two genotypes of OI-modeled mice for two weeks. The therapeutic efficacy of the treatment was evaluated two weeks following the final administration. The findings of the study suggest that the rBMP-2/SF spheres exhibited a sustained release of rBMP-2 for a minimum of 15 days *in vitro*. The osteoinductive activity of rBMP-2/SF particles was found to be well-maintained by the released factor. As compared to the treatment with blank SF spheres or rBMP-2 alone, the administration of rBMP-2/SF microspheres resulted in a notable enhancement of femoral microstructure and bone formation in OI-modeled mice. This research may contribute to the development of new strategies and therapeutic methods for the management of OI.

## MATERIAL AND METHODS

### Mice models

In this investigation, two genotypes of OI-modeled mice with mild disease phenotypes were utilized. The wild-type (WT) C57BL/6 mice were purchased from the Laboratory Animal Center of the Academy of Military Medical Science (Beijing, China). B6C3Fe a/a-$Col1a2^{oim/}$J mice (#001815) bought from Jackson Laboratory (Bar Harbor, ME, USA) were backcrossed with WT C57BL/6 mice for at least 11 generations. The heterozygotes were marked as $Col1a2^{oim/+}$ mice. Another genotype of OI mice was $Col1a1^{+/-365}$ which mimics OI type I (*Liu et al., 2019*). $Col1a1^{+/-365}$ and $Col1a2^{oim/+}$ mice were bred and identified following previously described procedures (*Liu et al., 2019*; *Carleton et al., 2008*)). All mice on a C57BL/6 background were bred under specific pathogen-free (SPF) conditions and the protocols of the study were approved by the Animal Care and Use Committee of Tianjin Medical University (TMUaMEC 2017012). This research employed only male mice.

### rBMP-2-loaded SF microspheres preparation and morphologically characterization

The lyophilized SF microspheres were bought from Suzhou Simatech Biotechnology Co., Ltd (China). A suspension of 2 mg/mL was prepared by dissolving the microspheres in a PBS solution using ultrasound. 0.5 $\mu$g of recombinant BMP-2 (rBMP-2; Peprotech, Cranbury, NJ, USA) was added into one mL of microsphere solution to produce a 0.5 $\mu$g rBMP-2/mL fibroin mixture, which was subsequently incubated for 30 min at 4 °C. Subsequently, the rBMP-2/SF microspheres were acquired through an overnight lyophilization process. The composite microspheres were sterilized by radiation. Both blank SF and rBMP-2/SF

microspheres were dissolved in PBS to prepare a 0.5 mg/mL solution in the following experiments.

The surface morphology of unloaded and loaded microspheres dissolved in PBS buffer was characterized by scanning electron microscopy (SEM). The solution of two kinds of spheres was centrifuged (1500 rpm/min, 5 min at 4 °C) and the supernatants were discarded. Then precipitates were resuspended in one mL sterile water separately. The SEM analysis was performed on gold-coated air-dried samples using a conventional SE2 detector (accelerating voltage of 2.00 kV) by GeminiSEM 300 microscope (Carl Zeiss Microscopy S.L., Oberkochen, Germany). The particle size of spheres was measured by ImageJ software.

## Adipose-derived mesenchymal stem cells (ADSCs) isolation, culture, and identification

ADSCs were used for testing cytotoxicity and the osteogenic activity of rBMP-2 released from the loaded spheres. ADSCs were isolated from 4-week-old WT and OI-modeled mice, following the methodology outlined in our previous publication (*Liu et al., 2021*). Briefly, the epididymis and groin adipose tissue were obtained and subjected to digestion using 0.2% type I collagenase (Sigma-Aldrich, USA). Subsequently, the cells underwent filtration and were cultured in $\alpha$MEM medium (Gibco, Billings, MT, USA), which was supplemented with 15% fetal bovine serum (FBS; Gibco) and 1% penicillin and streptomycin (PS; Gibco). Cells were passaged once the confluence reached 80%–90%. The three types of ADSCs were designated as ADSCs$^{WT}$, ADSCs$^{+/-365}$, and ADSCs$^{oim/+}$, respectively. Passage 3 (P3) generation of ADSCs after identification was used in this investigation.

P3 generation of ADSCs was identified *via* the detection of surface markers by flow cytometry (FCM) and differentiation ability assay. The cells were subjected to incubation with distinct antibodies, namely phycoerythrin (PE) conjugated-CD29 antibody, PE conjugated-CD44 antibody, PE conjugated-Sca1 antibody, fluorescein isothiocyanate (FITC) conjugated-CD45 antibody, and PerCP conjugated-CD90 antibody, all of which were obtained from eBioscience, USA. The incubation was carried out for 30 min at 4 °C. The detection of positive cells was performed using FCM. To analyze the data, FlowJo (Version 7.6.1) was employed. To evaluate the differentiation potential, ADSCs were cultivated for 14 days in the osteogenic, adipogenic, and chondrogenic induction medium (Cyagen, Santa Clara, CA, USA). Alkaline phosphatase (ALP; Solarbio, Beijing, China), Oil Red O (Solarbio), and Alcian blue (Solarbio) staining were used to determine the results.

## Cell viability assay

The cell viability assay using 3-(4, 5-Dimethylthiazol-2-yl)-2, 5-diphenyltetrazolium bromide (MTT) method (*Ghasemi et al., 2023*) was used to determine the cytotoxicity of SF microspheres. ADSCs were seeded in 96-well plates at a concentration of 7000 cells/well and cultured in a 37 °C, 5% $CO_2$ incubator for 24 h. The supernatant was then replaced with 200 µL of medium supplemented with 0, 0.5, 1, 2.5, 5, 25, 50, and 100 µg of SF microsphere. A total of 20 µL of MTT reagent (Solarbio) was added to each well after a 24-hour incubation period, and the cells were then continuously incubated for 4 h. The medium was then taken out, and 150 µL of DMSO (Solarbio) was used per well to dissolve

**Table 1  Primer sequences used for real time PCR assay.**

| Gene | Forward (5′-3′) | Reverse (5′-3′) | Length (bp) |
|---|---|---|---|
| *Alp* | CACGGCGTCCATGAGCAGAAC | CAGGCACAGTGGTCAAGGTTGG | 83 bp |
| *Col1a1* | GCTCCTCTTAGGGGCCACT | ATTGGGGACCCTTAGGCCAT | 91 bp |
| *Runx2* | GCAGCAGCAGCAGCAGGAG | GCACGGAGCACAGGAAGTTGG | 182 bp |
| *Bglap* | CTGACCTCACAGATGCCAAGCC | CATACTGGTCTGATAGCTCGTCACAAG | 192 bp |
| *Gapdh* | CATCACTGCCACCCAGAAGACTG | ATGCCAGTGAGCTTCCCGTTCAG | 153 bp |

the crystals. The optical density (OD) value at 490 nm was determined using the Microplate Reader (Bio-Rad, Hercules, CA, USA).

### *In vitro* release study

To study the release behavior, a mixture was prepared by immersing 0.3 mg rBMP-2/SF microspheres containing 0.075 µg rBMP-2, in three mL of PBS. The suspension was subjected to agitation while being placed in a water bath maintained at a temperature of 37 °C. A total of 100 µL of supernatant was collected at 1, 3, 6, 12, 24, 48, and 72 h, and then every three days for the next 30 days. The samples were centrifuged (13,000 rpm, 10 min), filtered by 0.22 µm membranes, and stored at −20 °C until the concentration was measured. After each sampling, an equivalent volume of PBS was added. The enzyme-linked immunosorbent assay (ELISA) kit (R&D Systems, Minneapolis, MN, USA) was utilized to detect the concentration of rBMP-2 in all samples following the manufacturer's instructions. All operational steps are completed under sterile conditions to avoid contamination.

### The biological activity of rBMP-2/SF microspheres by real-time quantitative polymerase chain reaction (qPCR)

To identify the osteoinductive activity of factors released from rBMP-2/SF microspheres, P3 ADSCs from OI-modeled mice were co-cultured with 20 µL of blank SF or rBMP-2/SF microspheres for 7 days. The medium was replaced on the third day and supplemented with an equivalent dosage of microspheres. Defective and wild-type ADSCs under normal culture served as control cells. ADSCs$^{WT}$ under normal culture conditions acted as the control. The extraction of total RNA from ADSCs was performed utilizing Trizol reagent (Invitrogen, USA) following the manufacturer's instructions. Reverse Transcription of mRNA to cDNA was performed using HiScript III RT SuperMix for qPCR (Vazyme, Nanjing, China). Real-time qPCR was performed using AceQ qPCR SYBR Green Master Mix (Vazyme). The procedures were established using the protocols reported in a prior report (*Liu et al., 2019*). The mRNA expression levels of several osteogenesis-related genes were normalized to *Glyceraldehyde-3-phosphate dehydrogenase (Gapdh)*. The primers given in Table 1 were produced by Sangon Biotech Co., Ltd (Shanghai, China).

### Animal studies

A total of 6 WT and 48 OI modeled mice, each 4 weeks old, were divided into the following groups: (1) WT + PBS group ($n = 6$) ; (2) *Col1a1$^{+/−365}$/ Col1a2$^{oim/+}$* + PBS group ($n = 6$); (3) *Col1a1$^{+/−365}$/ Col1a2$^{oim/+}$*+ SF group (for each genotype $n = 6$); (4) *Col1a1$^{+/−365}$/ Col1a2$^{oim/+}$*+ rBMP-2 group (for each genotype $n = 6$); (5) *Col1a1$^{+/−365}$/ Col1a2$^{oim/+}$*+

rBMP-2/SF group (for each genotype $n = 6$). Each mouse in the study received a single daily injection of PBS, rBMP-2 solution (0.5 µg/mL in PBS), SF microsphere solution, rBMP-2/SF microsphere solution, or rBMP-2 solution into the left femur. The right side femur received no medical treatment. The course of treatment lasted for two weeks. All mice were euthanized two weeks following the last injection, and the femurs were collected for later analysis.

## Micro-computed tomography (Micro-CT)

The micro-CT system (Scanco µCT-40; Bruttisellen, Zurich, Switzerland) was utilized following the instructions provided, to identify the femoral microstructure post-treatment by rBMP-2/SF microspheres. The attached software (CTAn and DataViewer) performed a three-dimensional analysis and calculation on scanned slice data. The region of interest (ROI) was selected from the proximal-distal growth plate, where the epiphyseal cap disappeared and continued for 200 slices to the end of the femur. The trabecular bone parameters assessed in this study included bone volume / total volume (BV/TV), trabecular number (Tb.N), trabecular thickness (Tb.Th), and trabecular spacing (Tb.Sp). Cortical bone parameters at mid-diaphysis were assessed through the measurement of cortical thickness (ct.Th) and cross-sectional area (CSA).

## Histological and immunohistochemical (IHC) staining

The femur samples were first fixed for three days in a 4% formaldehyde buffer, then decalcified for 21 days in a 10% EDTA buffer. The samples were then sectioned into tissue slices with a thickness of 5 µm and immersed in paraffin. Hematoxylin and eosin (H&E) staining (Solarbio) was used following the established methodology to evaluate the histological morphology.

IHC staining of Bone $\gamma$-carboxyglutamate protein (Bglap) was carried out for *in vivo* quantification of osteoblastogenesis The sections were dewaxed, hydrated, and subjected to antigen retrieval by microwave heating method at 96 °C for 30 min. The sections were then subjected to treatment with 0.1% Triton X-100 for 30 min, followed by blocking with 5% BSA for another 30 min at room temperature after cooling. The samples were treated overnight at 4 °C with rabbit-derived anti- Bglap (1:200; Bioss, China). HRP-conjugated goat anti-rabbit IgG (1:500; Bioss, Beijing, China) was employed as secondary antibodies the next day. Positive signals were detected through the utilization of the universal DAB color development kit (Beyotime, Jiangsu, China). The integral optical density (IOD) of Bglap positive signals across all sections was determined by Image-Pro Plus software (Media Cybernetics, Rockville, MD, USA). All the results were obtained from a minimum of six randomly selected fields for each sample.

## Statistical analysis

The statistical analysis of the data was performed using SPSS 17.0 software (SPSS Inc., Chicago, IL, USA), and the results were presented as mean ± SD. An ordinary one-way ANOVA of variance was used when more than two groups were compared. $P < 0.05$ was considered statistically significant.

# RESULTS

## The rBMP-2/SF microspheres sustainably released rBMP-2 over 15 days

The surface morphology of unloaded and loaded SF microspheres was observed through Scanning electron microscopy as depicted in Fig. 1A. The sphere solutions have been stored at −20 °C for a long term which might affect the morphology of microspheres. The SEM images showed that most of the unloaded and loaded spheres were spherical with rough surfaces (Fig. 1A). Some spheres manifested damaged surface morphology (Fig. 1A), which might be related to their degradation following storage in PBS. The particle size of blank and loaded spheres was 26.16 ± 10 μm and 26.42 ± 9.28 μm respectively (Fig. 1B). Overall, there was no significant difference in surface morphology and size between the two kinds of microspheres. The MTT assay was utilized to determine the cytotoxicity of SF microspheres. The results indicated that the introduction of SF particles ranging from 0.5 μg to 100 μg did not have any significant impact on the cell viability, as depicted in Fig. 1C. This observation indicates that the SF particles exhibit favorable biocompatibility. The results of the release experiment suggested that the SF microspheres loaded with rBMP-2 exhibited sustained release of the rBMP-2 for a minimum duration of 15 days *in vitro* (Fig. 1D). The cumulative release curve shown in Fig. 1D indicated that the release pattern of rBMP-2 exhibited an initial rapid release within the first three days (22.15 ± 2.88% of the loaded factor), followed by a transition to a slower and more consistent release, that persisted until the 15th day. The cumulatively released rBMP-2 did not change a lot post 15th day. After 15 days, 28.72 ± 0.96% of loaded rBMP-2 was released.

## Identification of ADSCs

The results depicted in Fig. 2A indicate that the P3 ADSCs exhibited a notable expression of CD44, CD29, CD90, and Sca1 while demonstrating minimal expression of CD45. These findings are in accordance with the typical features of mesenchymal stem cells (MSCs). The induction differentiation assay further confirmed their osteogenic, adipogenic, and chondrogenic differentiation capacities (Fig. 2B).

## The rBMP-2/SF microspheres promoted the osteogenic differentiation of ADSCs derived from OI-modeled mice

The bioactivity of rBMP-2 released from rBMP-2/SF microspheres was determined by analyzing the osteogenesis-related gene expression of mutant ADSCs cultured with rBMP-2/SF microspheres for 7 days. qPCR was employed to detect the mRNA expression levels of *Alp, Col1a1, Runx2*, and *Bglap* in stem cells. The results confirmed that the mutant ADSCs exhibited a reduced expression of the aforementioned genes in comparison to the normal stem cells (Fig. 3). Compared to untreated mutant cells, the transcription of these genes was significantly upregulated in defective stem cells following coculturing with rBMP-2/SF microparticles instead of blank SF spheres (Fig. 3). These results demonstrated that rBMP-2 released from rBMP-2/SF microspheres exhibited effective osteoinductive activity.

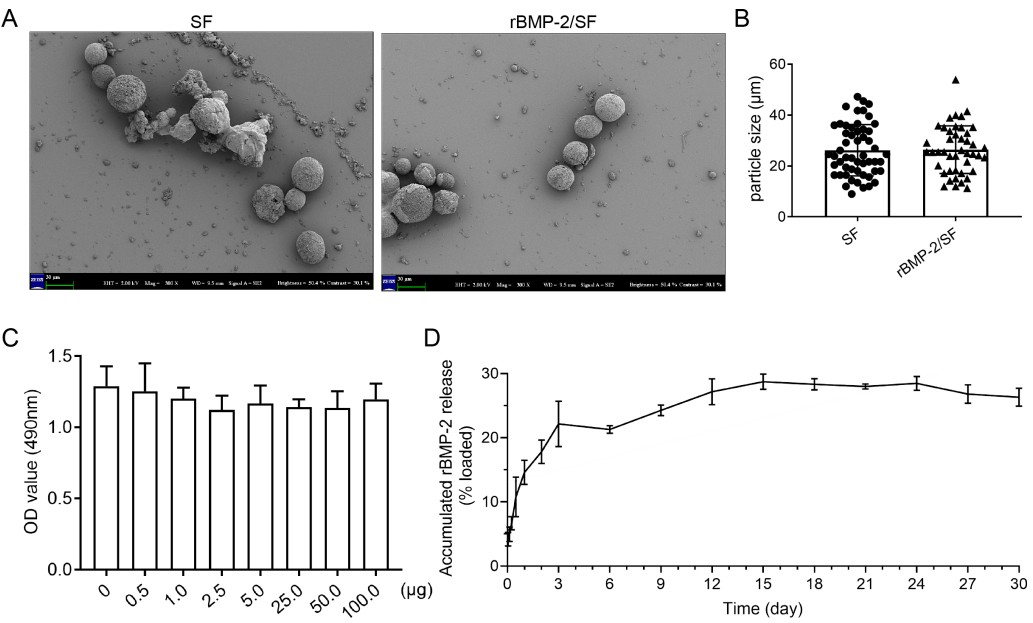

**Figure 1** **The characteristics of rBMP-2/SF microspheres.** (A) The representative SEM images of blank SF and rBMP-2/SF microspheres previously dissolved in PBS. Most of the spheres were spherical particles with unsmooth surface and some showed collapsed surface; (B) The particle size of the two kinds of microspheres; (C) The biocompatibility of bland SF microspheres estimated by MTT assay. Addition of SF microspheres ranging from 0.5 μg to 100 μg did not affect the cell viability; (D) The release curve of rBMP-2/SF microspheres for 30 days. The release pattern of rBMP-2 exhibited an initial rapid release within the first three days ($22.15 \pm 2.88\%$ of the loaded factor), followed by a transition to a slower and more consistent release rate that persisted until the 15th day.

## Infusion of rBMP-2/SF microspheres improved femoral microstructure of OI-modeled mice

The structural parameters of murine femurs were analyzed using Micro-CT. Figure 4 depicts the reconstructed 3D images of each group of mice. The $Col1a1^{+/365}$ mice with a null $Col1a1$ allele closely mimic the OI type I (*Liu et al., 2019*). Another OI mouse model, $Col1a2^{oim/+}$, with a mutation(c.3978del) in the $Col1a2$ gene, resembles OI type IV (*Saban et al., 1996*). In accordance with earlier findings (*Liu et al., 2021*), it was observed that both PBS-treated OI-modeled mice exhibited significant loss of trabecular and cortical bone in comparison to WT mice treated with PBS (Fig. 5). The injection of blank SF spheres did not affect any skeletal parameters (Fig. 5). Compared to PBS-treated mice of the same genotype, a few parameters improved significantly in OI-modeled mice that received rBMP-2 factor alone (Figs. 5A, 5C, 5E–5H and 5K). Both OI-modeled mice treated with rBMP-2/SF microspheres demonstrated substantially enhanced trabecular and cortical performance compared to mice of the same genotype treated with PBS only (Fig. 5).

## Infusion of rBMP-2SF/ microspheres induced bone formation of OI-modeled mice

H&E staining showed the microanatomy of the bone in each mouse. Compared to PBS-treated WT mice, both $Col1a1^{+/-365}$ and $Col1a2^{oim/+}$ mice appeared to have fewer and

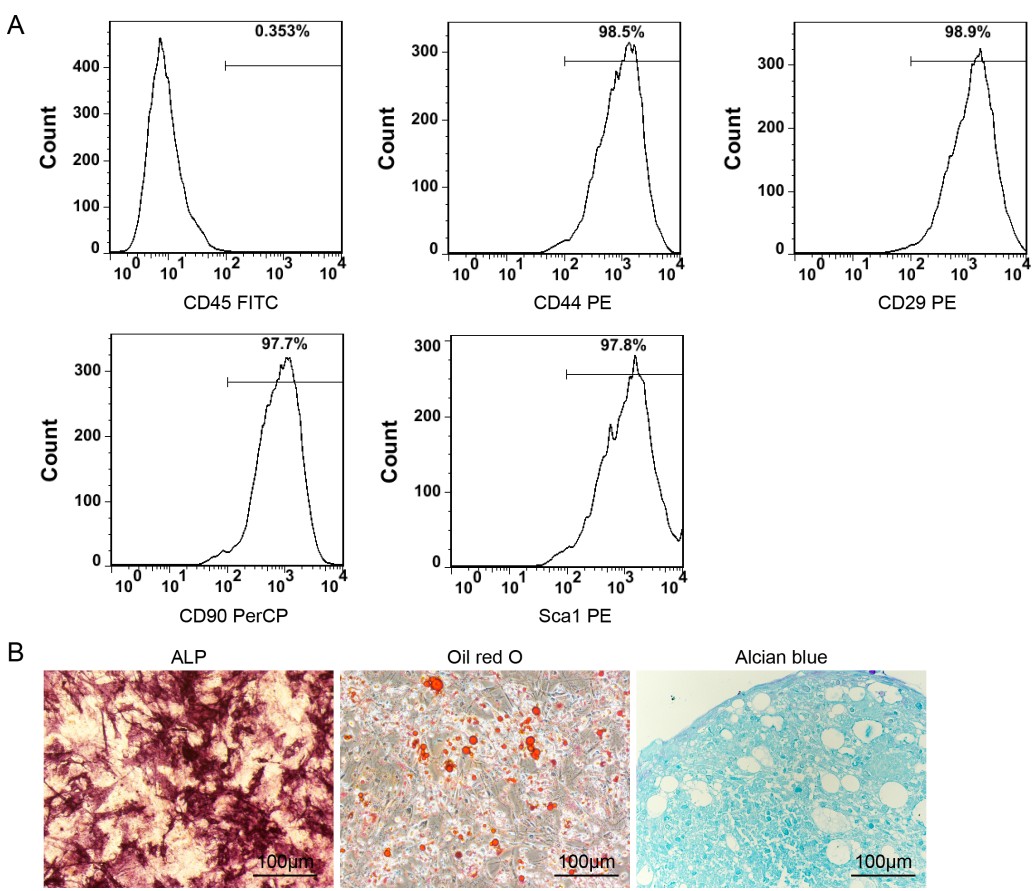

**Figure 2** **Identification of ADSCs.** (A) The surface markers detection of ADSCs by FCM. P3 ADSCs expressed high level of CD44, CD29, CD90 and Sca1 but almost none of CD45; (B) the ALP, Oil Red O and Alcian blue staining images of ADSCs after 14 days of osteogenic, adipogenic, and chondrogenic differentiation induction.

smaller trabeculae, particularly at the lower edge of the growth plate (Figs. 6A and 6B). Quantitative results of bone area (B.Ar) and bone perimeter (B.Pm) showed no significant changes in the histological performance of blank SF microspheres or rBMP-2 treated-OI mice when compared to the same genotype treated with PBS (Figs. 6C–6F). Mice treated with rBMP-2/SF microspheres had significantly larger B.Ar and B.Pm than the same genotype treated with PBS, SF particles, or rBMP-2 (Figs. 6C–6F). These results indicated that rBMP-2/SF spheres effectively promoted bone formation in the two genotypes of OI mice.

## Infusion of rBMP-2/SF microspheres promoted Bglap expression of OI-modeled mice

Bglap, also known as osteocalcin, is primarily produced by osteoblasts and regulates bone calcium metabolism (*Komori, 2020*). The Bglap expression can reflect the extent of bone formation. Immunohistochemical staining of Bglap demonstrated that OI mice had a much lower level of Bglap in the femurs than PBS-treated WT mice (Fig. 7). Blank SF particles

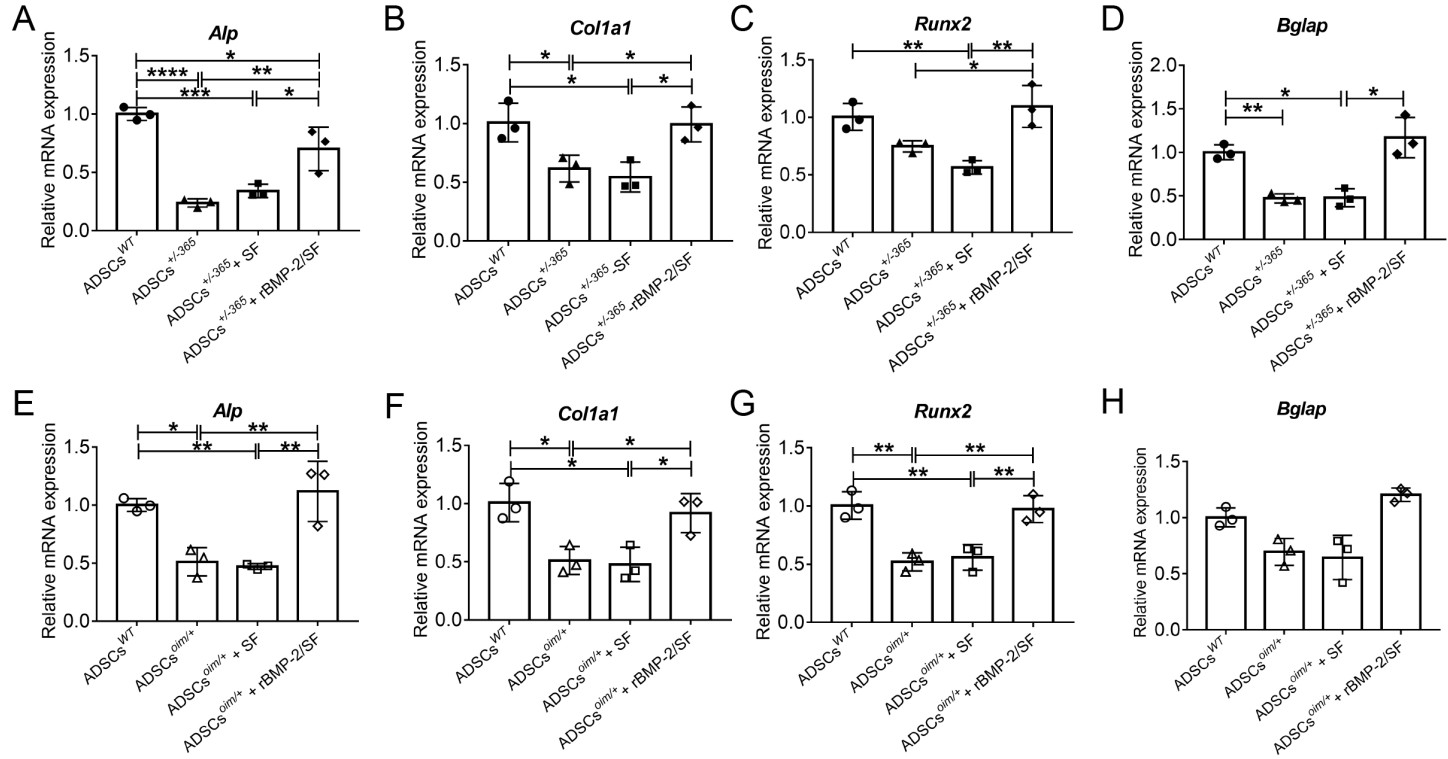

**Figure 3** **The mRNA level of genes involved in osteogenic differentiation of rBMP-2/SF microspheres treated ADSCs.** Relative mRNA expression of osteogenesis related genes including *Alp* (A, E), *Col1a1* (B, F), *Runx2* (C, G) and *Bglap* (D, H) of defective ADSCs coculture with SF or rBMP-2/SF microspheres for 7 days. Wild type and mutant cells under normal culture served as controls. The factor released from rBMP-2/SF microspheres promoted transcription of these genes. *$P < 0.05$, **$P < 0.01$, ***$P < 0.001$, ****$P < 0.0001$.

or rBMP-2 treatment did not increase Bglap expression in OI mice (Fig. 7). Infusion of rBMP-2/SF microspheres significantly increased Bglap expression in comparison to the same genotype treated with PBS, SF particles, or rBMP-2 (Figs. 7C and 7D). These results indicated that rBMP-2/SF microspheres could promote bone metabolism and bone formation in both OI mice.

## DISCUSSION

OI is an inherited disorder for which a cure has unfortunately not yet been identified. Current clinical treatments attempt to increase bone mass while decreasing the occurrence of bone fractures (*Morello, 2018*). OI is also categorized as a congenital disease characterized by substantial genetic and phenotypic heterogeneity. Although more than 21 causal genes are associated with the disease, hereditary deficiency in type I collagen production is the main cause of OI (*Fratzl-Zelman et al., 2015*). It has been reported that reduced bone formation and excessive bone resorption contribute significantly to the brittle bones of collagen-related OI, which are the primary treatment targets for OI (*Rossi, Lee & Marom, 2019*). Stem cell therapy has shown promising prospects in the treatment of OI as a result of its beneficial effects on bone formation (*Gotherstrom & Walther-Jallow, 2020*). The

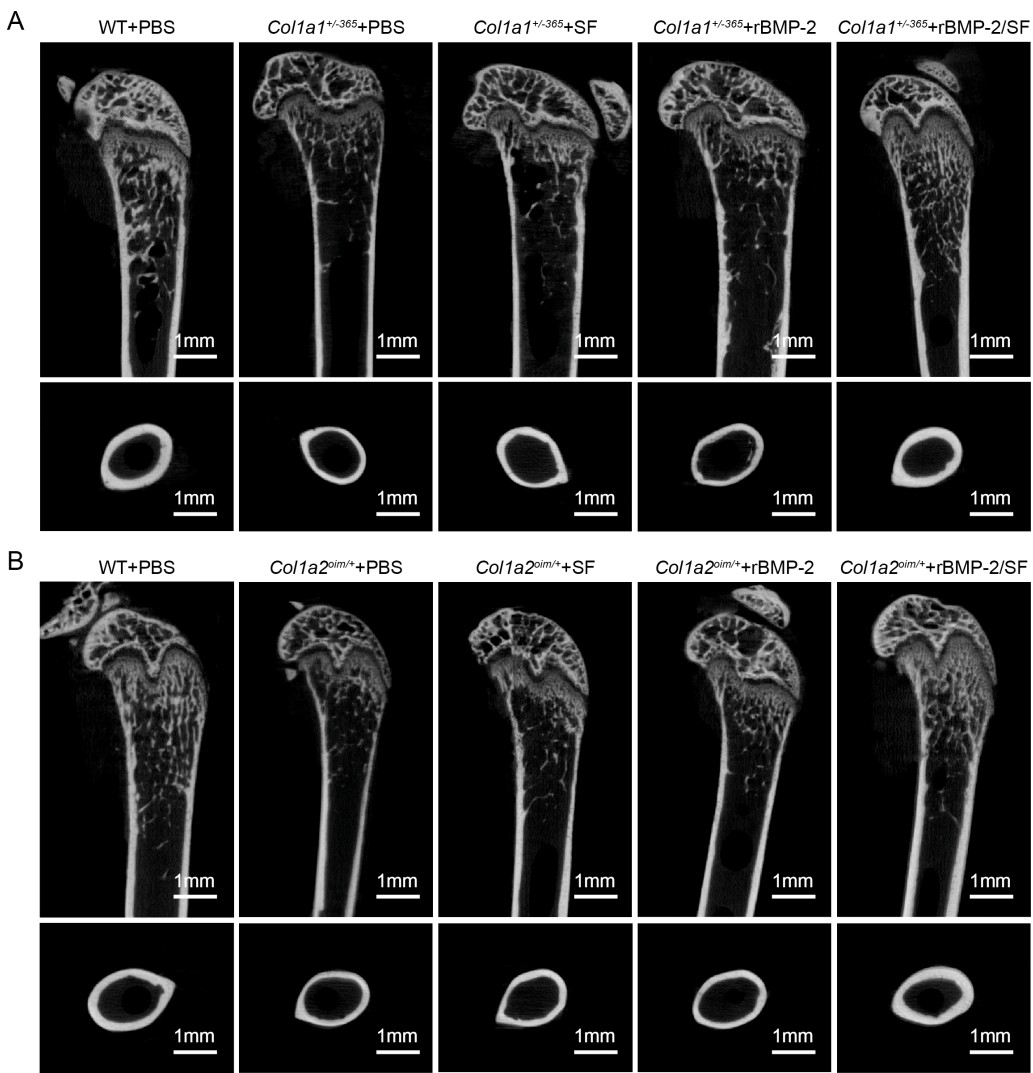

**Figure 4** **(A–B) Femoral Micro-CT detection 2 weeks after the final injection.** The representative reconstructed 3D images of each group 2 weeks after the final injection.

cumbersome operation of cell therapy may hinder its widespread application. Drug therapy is relatively more practical and convenient in clinical practice. Multiple anti-resorptive and anabolic agents have improved the phenotype of OI bones (*Datta, Vila & Tuck, 2021*; *Shanas et al., 2022*). However, the length of these medications' half-life frequently results in undesirable side effects or inconveniences. For exogenous drugs with rapid clearance and degradation, local and sustained delivery is an effective method.

As a powerful cytokine that can promote bone formation, BMP-2 has tremendous application potential in the healing of large bone defects and the enhancement of the rate of bone formation. BMP-2 has been approved by the U.S. Food and Drug Administration (FDA) for bone regenerative therapy since 2002, suggesting its commercial availability for clinical use. However, there are concerns about the dosage of BMP-2. According to studies,

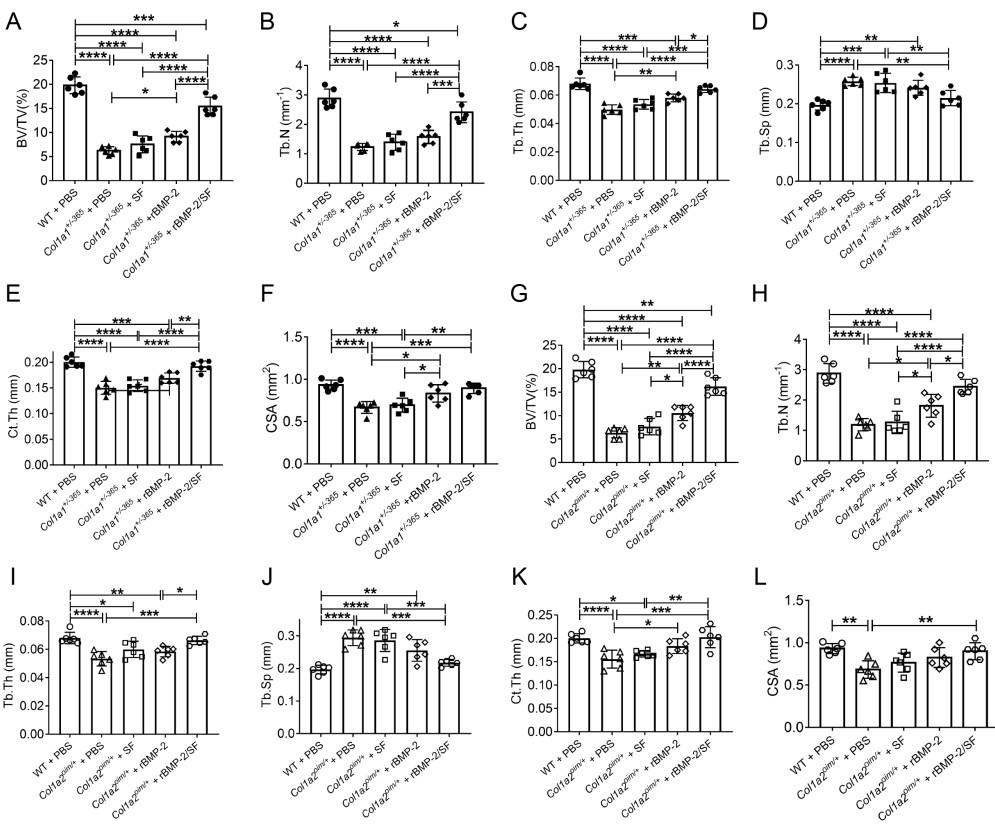

**Figure 5** **Quantitative analysis of femoral Micro-CT detection 2 weeks after the final injection.** Quantitative analysis of trabecular and cortical parameters of each group including bone volume/tissue volume ratio (BV/TV) (A, G), trabecular number (Tb.N) (B, H), trabecular thickness (Tb.Th) (C, I), trabecular separation (Tb.Sp) (D, J), cortical thickness (Ct.Th) (E, K) and total cross-sectional area (CSA) (F, L). Infusion of rBMP-2/SF microspheres, as opposed to blank SF spheres or rBMP-2 monotherapy, improved the femoral microstructure of OI mice. *$P < 0.05$, **$P < 0.01$, ***$P < 0.001$, ****$P < 0.0001$.

a low concentration of BMP-2 may not deliver the desired therapeutic effect, whereas a supraphysiological dosage of BMP-2 may cause unpredictable side effects, such as ectopic osteogenesis and even tumors (*Hankenson, Gagne & Shaughnessy, 2015*; *Li et al., 2022*). Most researchers adopt sustain-release technology to limit the release of BMP-2 within the body and to delay its degradation. A few studies have used BMP-2, in a slow-release manner, to treat mice modeled with OI. *Cheng et al. (2019)* developed a sucrose acetate isobutyrate (SAIB) solution as an effective carrier to deliver BMP-2 into the reamed tibias through intraosseous injection, resulting in an increase in cortical bone thickness in a type of OI mice (*Col1a2$^{+/G610C}$*). In another study, a small cubed acellular collagen sponge was used as a BMP-2 sustained-release technique to treat tibial fractures in *Col1a2$^{+/G610C}$* mice, increasing trabecular bone (*O'Donohue et al., 2022*).

In addition to the aforementioned sustained-release techniques, microparticulate systems have also been extensively employed. Silk fibroin (SF) microspheres are natural proteins derived from silk that have excellent histocompatibility and biodegradability

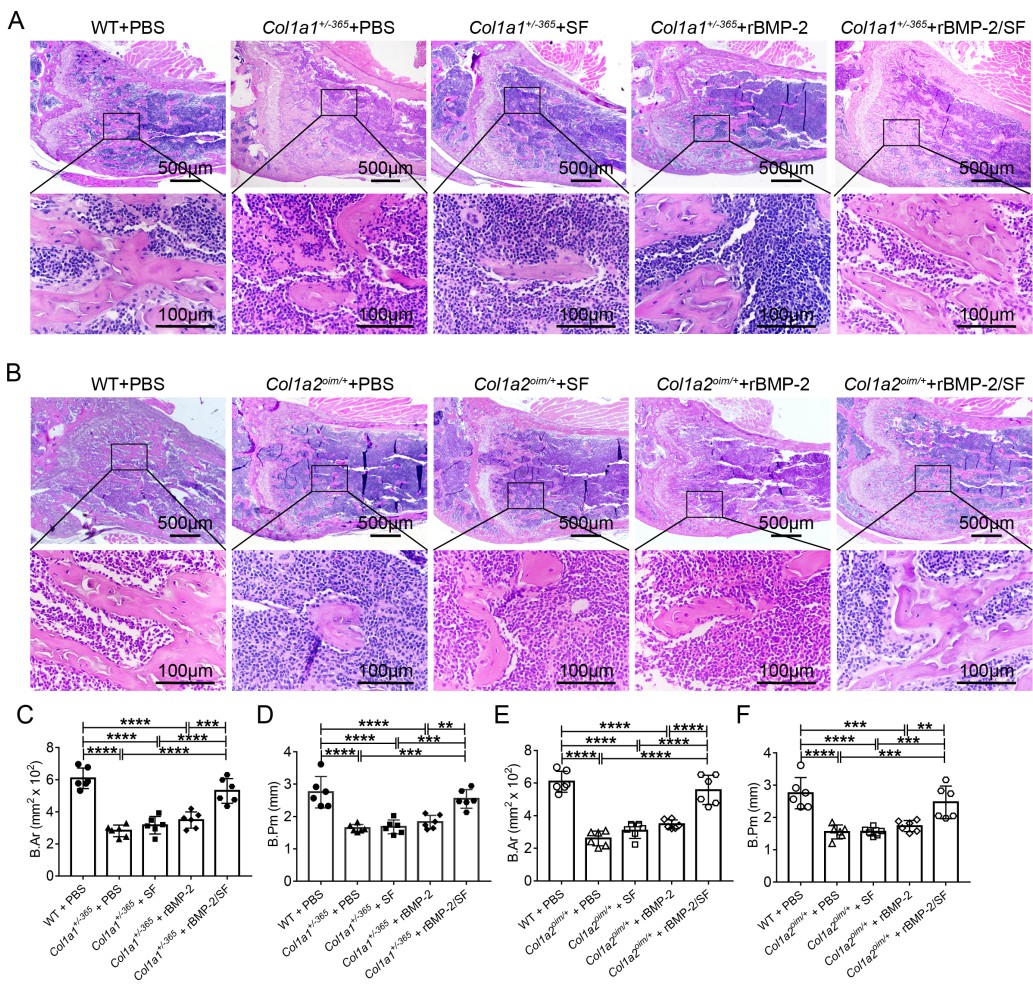

**Figure 6 Histological staining of femurs 2 weeks after the final injection.** (A, B) The representative H&E staining images of each group 2 weeks after the final injection; (C–F). Quantification analysis of the bone area (B.Ar) (C, E) and bone perimeter (B.Pm) (D, F). Infusion of rBMP-2/SF microspheres instead of SF spheres or rBMP-2 promoted bone formation of OI mice. **$P < 0.01$, ***$P < 0.001$, ****$P < 0.0001$.

(*Kundu et al., 2013*). As drug carriers, SF microparticles can adsorb or encapsulate growth factors and small molecular drugs and give an almost steady release of drugs locally for a longer duration of time (*Farokhi et al., 2020*; *DeBari et al., 2021*), rendering them widely useful in bone regeneration and repair (*Wu et al., 2022*). However, the SF delivery system has not been used in the treatment of OI. In the present investigation, biocompatible SF microspheres (Figs. 1A–1C) were utilized to physically adsorb recombinant BMP-2 (rBMP-2). The rBMP-2-loaded SF (rBMP-2/SF) microspheres exhibited an initial rapid release within the first three days (22.15 ± 2.88% of the loaded factor), followed by a transition to a slower and more consistent release rate that persisted until the 15th day (Fig. 1D). After 15 days, 28.72 ± 0.96% of loaded rBMP-2 was cumulatively released. The osteogenesis-related gene expression of ADSCs generated from two OI-modeled mice was significantly upregulated after coculturing with rBMP-2/SF microparticles (Figs. 2 and 3),

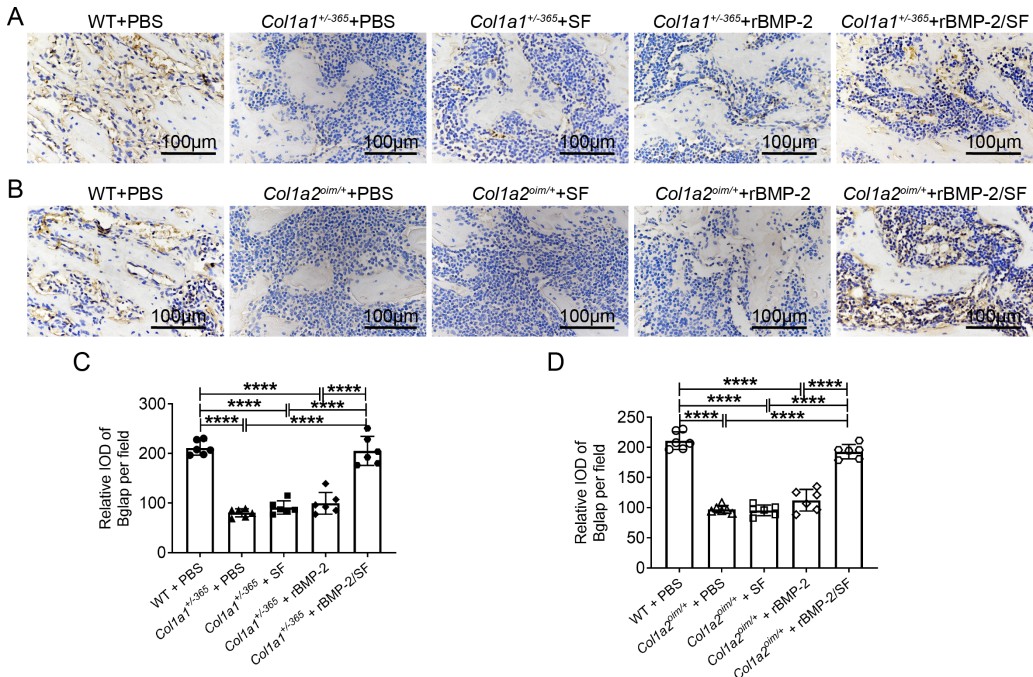

**Figure 7** **Immunohistochemical staining of Bglap on femur sections 2 weeks after the final injection.** (A, B) The representative immunohistochemical staining images of Bglap of each group 2 weeks after the final injection; (C, D) Statistics of integrated optical density (IOD) of Bglap per field. Infusion of rBMP-2/SF microspheres instead of SF spheres or rBMP-2 promoted Bglap expression of OI mice. ****$P <$ 0.0001.

demonstrating the good bioactivity of factors released from the composite spheres. The rBMP-2/SF microspheres were then infused into the bone marrow cavity of the femoral bone of the OI-modeled mice, allowing a slow and continuous release of the factor locally. The same dose of blank SF particles or rBMP-2 factor alone treated the OI mice serving as control groups. Figures 4–7 demonstrates that rBMP-2/SF microspheres substantially improved the femoral microstructure and promoted bone formation in OI-model mice when compared to blank SF spheres or rBMP-2 alone. The failure of rBMP-2 alone to improve the skeletal phenotype of OI bones could be attributed to its rapid clearance and degradation *in vivo*.

The special porous network membrane structure endows silk fibroin with excellent adsorption and sustained-release functions, which can delay the release rate of drugs thereby rendering it useful as good sustained-release agent. SF encapsulation can regulate the release time and degradation rate of BMP-2 through self-degradation.

It is noteworthy to mention that the current study did not observe any apparent instances of ectopic osteogenesis, which could potentially be attributed to the relatively low dose of rBMP-2 administered and the localized sustained release treatment. However, systematic investigation on the safety and efficacy of rBMP-2/SF microspheres therapy for OI is still required. In addition, the sustained release of the adsorbed factor from the loaded microspheres lasted for about ten days in the *in vitro* release experiment. The modification

of SF materials and/or the improvement of preparation methods for loaded microspheres may contribute to the extension of sustained-release duration, which also requires further research.

## CONCLUSION

In conclusion, the present study has demonstrated the utilization of SF microspheres as carriers for the controlled and sustained release of rBMP-2 *in vivo*, to treat two types of OI mice. The rBMP-2/SF microspheres prepared by surface adsorption retained the bioactivity of rBMP-2 and released the factor for at least 15 days *in vitro*. The administration of rBMP-2/SF microspheres *via* femoral bone marrow cavity infusion resulted in the restoration of the bone phenotype in both OI-modeled mice. This study may yield new therapeutic strategies and methods for the treatment of OI.

### Funding

This study was supported by research grants from the National Key R&D Program of China (2017YFC1001904) and National Natural Science Foundation of China (82272447). The funders had no role in study design, data collection and analysis, decision to publish, or preparation of the manuscript.

### Grant Disclosures

The following grant information was disclosed by the authors:
National Key R&D Program of China: 2017YFC1001904.
National Natural Science Foundation of China: 82272447.

### Competing Interests

The authors declare there are no competing interests.

### Author Contributions

- Ting Fu conceived and designed the experiments, performed the experiments, analyzed the data, prepared figures and/or tables, authored or reviewed drafts of the article, and approved the final draft.
- Yi Liu conceived and designed the experiments, prepared figures and/or tables, authored or reviewed drafts of the article, and approved the final draft.
- Zihan Wang performed the experiments, prepared figures and/or tables, authored or reviewed drafts of the article, and approved the final draft.
- Yaqing Jing analyzed the data, prepared figures and/or tables, contributed reagents and materials, and approved the final draft.
- Yuxia Zhao analyzed the data, prepared figures and/or tables, contributed reagents and materials, and approved the final draft.
- Chenyi Shao analyzed the data, prepared figures and/or tables, contributed reagents and materials, and approved the final draft.

- Zhe Lv analyzed the data, prepared figures and/or tables, contributed reagents and materials, and approved the final draft.
- Guang Li conceived and designed the experiments, prepared figures and/or tables, authored or reviewed drafts of the article, and approved the final draft.

## Data Availability

The analysis of micro-CT and staining in this study are available in the Supplemental Files. The 54 micro-CT MorphoSource DOIs are available in the Supplemental File.

## Supplemental Information

Supplemental information for this article can be found online at http://dx.doi.org/10.7717/peerj.16191#supplemental-information.

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
