# Peer review of "The recombinant BMP-2 loaded silk fibroin microspheres improved the bone phenotype of mild osteogenesis imperfecta mice"

_PeerJ, doi:10.7717/peerj.16191_

## Round 0.1 · original submission · Major Revisions

The manuscript has been assessed by three independent reviewers and I strongly suggest addressing the concerns raised by all three reviewers before your paper could be considered for publication.

1. Language revision is highly recommended for better understanding.
2. The authors need to reframe the abstract section concentrating mainly on research questions and the approach taken for the current study in order to bring a flow while reading.
3. More concentration is required on linking different experiments and rationale behind performing those experiments.
4. The figures need extensive reformatting and reorganization.
5. Extensive revision of methods section is highly recommended.
6. The authors are suggested to provide more validation and evidence for the following:
a) adsorption of rBMP-2 on SF microsphere
b) differentiation between loaded and unloaded SFs
c) sustained release of rBMP-2 until 30 days

Reviewer 1 ·

Basic reporting

no comment

Experimental design

1) In Figure 1C, the author shows the concentration of BMP-2 in PBS solution as a function of time, till 30 days after rBMP-2/SF was dissolved in PBS buffer. Based on the figure author suggested (Section 3.1), the release of BMP-2 significantly occurs in the first 3 days and remains relatively stable until the 30th day. However, it looks like the major release occurs in the first 3 days, and the curve plateaus after 10 days. Therefore the concentration of BMP-2 after 10 days is the same as the concentration after 30 days (within the experimental error limit).

So, the author’s suggestion that BMP-2 keep getting released for 30 days is inappropriate. I would suggest the author plot the relative change in concentration with respect to the concentration of BMP-2 on the previous day. This will help in rationalizing the percentage change in the concentration of BMP-2 released in the solution after 10 days of dissolution.

2) Did the authors try any other concentration for the preparation of BMP-2/SF Microspheres?? If so, did the dissolution kinetics of BMP-2 remained the same or change significantly?

3) I would suggest authors add a BMP-2 release profile at 4 C temperature to provide an idea of the stability of the microsphere at near ice temperature.

Validity of the findings

In Figure 3, authors have shown relative mRNA expression of osteogenesis genes and show that upon incubation with rBMP-2/SF microspheres, the expression of osteogenesis genes reverts back to WT. However, the error in the estimation of mRNA expression is quite high in the case of rBMP-2/SF microspheres, as compared to WT. It seems that the concentration range of mRNA in the case of rBMP-2/SF incubated conditions lies from WT to ADCs +/-365.

I would suggest the author perform one more set of experiments and average it out with current data to show the changes more cleanly.

·

Basic reporting

The introduction is self-explanatory.

Objectives, Material and methods, and Discussion are also clear.

Literature Review was sufficient.

The table is clear.

Sufficient raw data has been supplied.

The manuscript is clearly written in professional, unambiguous language.

Experimental design

No comment

Validity of the findings

No comment

Additional comments

The article discusses Osteogenesis Imperfecta (OI), a genetic disorder that affects bone mass and fragility, and is usually caused by mutations in the genes COL1A1 or COL1A2. Current clinical management of OI involves orthopedic surgery and anti-resorptive medication, but bone anabolic therapy may be more advantageous in decreasing fracture incidence. In this study the authors explore the use of recombinant BMP-2 (rBMP-2)-loaded silk fibroin (rBMP-2/SF) microspheres for sustained release and targeted delivery of osteogenic factors to improve bone formation in OI modeled mice. The results showed that rBMP-2/SF microspheres significantly improved the femoral microstructure and promoted bone formation, indicating a potential new therapeutic method for OI management. This is an important finding that will be of broad interest to the community.
There are, however, a few issues that should be addressed in the presentation and discussion of data, as outlined below.
While I don’t doubt that the authors are loading biocompatible (Fig. 1b) SF (Fig. 1a) with rBMP-2, which is released until 30th day (Fig. 1c), however it should be discussed how the authors examined the differences between loaded and unloaded SFs. To be precise, it would be interesting to have a statistical value on the percentage of loaded SFs over unloaded SFs.
In figure 1a, the authors have shown the spherical and unsmooth surface of SF brilliantly through SEM. To get a better understanding of BMP-2 release, it would be beneficial to see BMP-2 loaded SF, if possible in the similar manner.
In figure 1c, the release curve showed that rBMP-2 release transited to slow and relatively stable release after 3rd day, until the 30th day. Is there any specific justification for that? The authors should shed some light on future direction of more homogenous release of the drug in discussion paragraph.
In the title, it has been mentioned about improving ‘bone phenotype of mild Osteogenesis imperfecta mice’. In this work, OI mouse Col1a2oim/+ having a mutation (c.3978del) in Col1a2 gene has been used. Col1a2 gene resembles OI type IV. Cohen-Solal et. al. (Hum Genet, 1991,87(3):297-301) classified OI type IV as moderately severe. Please explain this disparity.
In figures 4 and 5, trabecular and cortical performance has been compared with respect to WT and rBMP-2/SF microsphere treatment showed exciting development. However, improvement in trabecular separation has different appearances with respect to WT, when compared between two different genotypes Col1a1 and Col1a2. How it can be explained?
Figure 6 shows, very convincing result on bone formation using rBMP-2/SF. Why the statistical deviation is higher for WT+PBS in bone area and bone perimeter when compared with both Col1a1 (Fig 6C, 6D) and Col1a2 (Fig 6E, 6F)?
In figure 7, immunohistochemical staining of BGLAP shows promotion of bone metabolism upon rBMP-2/SF treatment however statistics of integrated optical density for BGLAP of Col1a1 has highest deviation. What could be the reason for that?

·

Basic reporting

Regarding Grammar: I appreciate the authors for their efforts to put together a great piece of scientific research. However, to reach out to the international community and achieve the Journal’s objective, the entire manuscript requires a complete remodeling to improve the language and the grammar substantially. A few examples can be found here: Abstract: Lines 13-14 highlights problem 1, likewise lines 15-16 identifies problem 2. Each problem needs to be addressed with a specific solution. There should be a continuity and flow of sentences while addressing successive problems accompanied by possible solutions. Introduction: Line 35: ‘bone seriously affected in OI’ can be rephrased as ‘it more vulnerable in patients suffering from OI’. Line 35: As per Sillence’s. Line 55: correct the grammar ‘Osteogenic including’. Materials and methods: Line 119: ‘placed’ is not appropriate, can be replaced with ‘incubated’. These are some of the isolated examples, and in general the manuscript requires a complete reformat, which can help reach out to the larger international community.
Introduction clearly defines the OI, the causes, possible treatments, and limitations along with the author’s approach in addressing the problem through SF microspheres. Sufficient references have been provided in the introduction section.
The Results section lacks rationale for most of the experiments. The authors have not defined why they even performed a given experiment. 3.4 was an exception, the authors defined why they used micro-CT. Likewise, there should be a continuous flow in the results connecting from previous experiment and to subsequent experiments.
The figures are clear with elaborate description. However, a few changes may be necessary. Figure 1B: Use a gradient color from 0 to 100 µg. Figure 1C: X-axis should be ‘Time (days)’. Figure 2A: Describe X-axis in all the graphs.
I suggest the authors combine the images 4A and 5A as one figure representing Figure 4A and 4B. This will have all the reconstructed 3D images in place. Likewise, Figure 4B until 4G and Figure 5B until 5G, be clubbed as one Figure, as Figure 5A – 5L. This change would have a similar set of experiments in one place and be easy to navigate. Otherwise, the legends for 4 and 5 are the same, except for the difference in Genotype.

Experimental design

The overall experimental design is well structured and well within the scope of the journal. The authors raised concerns regarding the conventional therapeutic methods for treating OI, with a potentially possible therapeutic method through the sustained release of BMP-2 from SP microspheres. However, as previously stated, the language needs to be significantly improved to reach out to the broader community. Although SF microspheres have been previously demonstrated as potential therapeutic agents for treating certain bone defects, in this study authors have shown that this method can be further extended to possibly treat OI. This certainly narrows the knowledge gap by various means as to how SF microspheres can possibly be implemented for therapeutic purposes. I also believe that the methods are easily reproducible.

Validity of the findings

The authors have mentioned that the rBMP-2 has been loaded on SF microspheres via adsorption. However, the authors have not provided any substantial evidence. Authors may provide a SEM image of rBMP-2/SF or any other experimental data that validates the adsorption of rBMP-2 on SF microspheres.

Otherwise, the rest of the data appears good and reproducible. Authors have used various techniques like SEM, ELISA, flow cytometry and Micro-CT in this study, which is justified to address the objectives defined in the manuscript. Conclusion is well defined with their overall results, concerning the questions put forward in the introduction.

Additional comments

General comments:
Lines 27 & 28: Keywords: Sustained release. Authors have provided only four keywords. They may add additional keywords like Congenital disease and/or Micro-CT.
Line 119: it is mL, not mg
Lines 123 & 124: MTT – Define the full form of MTT. Also, provide a reference for this method.
Line 148: Glyceraldehyde – Check the spelling.
Line 164: ‘attached software’, it would be nice to provide the software name and its version.
Line 225: (Figure 4 A ---) Remove A.
Lines 229 – 238: Figure 6 E and 6 F are not mentioned in the manuscript.
Lines 241: Provide reference/s for Osteocalcin.
Line 286: Check the spelling of microspheres.

---

## Round 0.2 · Minor Revisions

I appreciate the effort taken by the authors to improve the overall quality of the manuscript. However, there are very minor errors which needs to be corrected as suggested by one of the reviewers. Hence, the authors are suggested to carry out the necessary corrections and submit the manuscript.

Reviewer 1 ·

Basic reporting

n/a

Experimental design

n/a

Validity of the findings

n/a

Additional comments

The authors have provided a satisfactory response to the queries raised. However, they do express their inability to perform a few suggested experiments, as they ran out of SF microspheres. Nevertheless, based on the logically sound response provided by them, I recommend the paper be published.

·

Basic reporting

No further comments.

Experimental design

No further comments.

Validity of the findings

No further comments.

Additional comments

No further comments.

I would like to thank the authors for addressing the comments, the addition of the extra detail is very welcome. I really enjoyed reading the manuscript:)

·

Basic reporting

Regarding Grammar: I appreciate the authors for their efforts to put together a great piece of scientific research. The grammar has been substantially improved from the previous version, however only a few grammatical errors need to be rectified. The authors have provided sufficient references and also updated a few more on the basis of previous suggestions. The figures and legends are well structured and defined. Importantly, the scientific data from this study agrees with the author’s hypothesis.

Experimental design

The overall experimental design is well structured and well within the scope of the journal. The authors raised concerns regarding the conventional therapeutic methods for treating OI, with a potentially possible therapeutic method through the sustained release of BMP-2 from SP microspheres. To address the manuscript’s main aims, the authors have designed and performed several experiments and validated the importance of sustained release of BMP-2. Although SF microspheres have been previously demonstrated as potential therapeutic agents for treating certain bone defects, in this study authors have shown that this method can be further extended to possibly treat OI. This certainly narrows the knowledge gap by various means as to how SF microspheres can possibly be implemented for therapeutic purposes. I strongly believe that these methods are standard and are easily reproducible.

Validity of the findings

In the previous review, the authors were requested to update the manuscript with a SEM image of rBMP-2/SF. The updated manuscript looks promising, and the various techniques used by the authors like the SEM, ELISA, flow cytometry and Micro-CT in this study, have synergistic conclusions that validate their original hypothesis. Conclusion section is well defined

Additional comments

Line 18: replace ‘experienced’ with ‘tested’
Line 41: peptides α1 (give space)
Line 43: fibrils of collagen (remove full stop)
Line 94: replace ‘yield’ with ‘contribute to’
Line 107: replace ‘experiment’ with ‘work’ or ‘research’
Line 174: correct the grammar, ‘as acted as the control’
Line 180: Glyceraldehyde – Check the spelling. This was not rectified in the previous review as well.
Lines 230-232: ‘Some spheres _____ wet state’. This statement is not appropriate. I request authors to rewrite this.
Lines 315-316: Correct the sentence, ‘However ___of BMP-2’ as ‘However there are concerns about the BMP-2 dosage.’
Lines 358-360: ‘Modification of _____ researches’. Kindly check the grammar.

---

## Round 0.3 · Minor Revisions

The authors have done a great job in revising the manuscript. However, few grammatical errors need to be corrected as suggested by one of the reviewers before publication.

**Language Note:** The Academic Editor has identified that the English language must be improved. PeerJ can provide language editing services - please contact us at [email protected] for pricing (be sure to provide your manuscript number and title). Alternatively, you should make your own arrangements to improve the language quality and provide details in your response letter. – PeerJ Staff

---

## Round 0.4 · accepted · Accept

The authors have corrected grammatical errors pointed out by the reviewer and is ready for publication.